# Safety and anti-tumour activity of the IgE antibody MOv18 in patients with advanced solid tumours expressing folate receptor-alpha: a phase I trial

James Spicer [1,2] ✉, Bristi Basu[3], Ana Montes[2], Udai Banerji [4], Rebecca Kristeleit [2], Rowan Miller [5], Gareth J. Veal[6], Christopher J. Corrigan [7], Stephen J. Till[7], Mariangela Figini [8], Silvana Canevari [9], Claire Barton[10,11], Paul Jones[10,16], Sarah Mellor[10], Simon Carroll[10], Chris Selkirk[10], George Nintos[2], Vineet Kwatra[2], Ionut-Gabriel Funingana [3], Gary Doherty [3], Hannah J. Gould [7,12], Giulia Pellizzari [13], Mano Nakamura [13], Kristina M. Ilieva[13], Atousa Khiabany[13], Chara Stavraka[1,2,13], Jitesh Chauhan[13], Cheryl Gillett[1,2,14], Sarah Pinder [1,2,14], Heather J. Bax [1,13], Debra H. Josephs[1,2,13] & Sophia N. Karagiannis [13,15]

All antibodies approved for cancer therapy are monoclonal IgGs but the biology of IgE, supported by comparative preclinical data, offers the potential for enhanced effector cell potency. Here we report a Phase I dose escalation trial (NCT02546921) with the primary objective of exploring the safety and tolerability of MOv18 IgE, a chimeric first-in-class IgE antibody, in patients with tumours expressing the relevant antigen, folate receptor-alpha. The trial incorporated skin prick and basophil activation tests (BAT) to select patients at lowest risk of allergic toxicity. Secondary objectives were exploration of anti-tumour activity, recommended Phase II dose, and pharmacokinetics. Dose escalation ranged from 70 µg–12 mg. The most common toxicity of MOv18 IgE is transient urticaria. A single patient experienced anaphylaxis, likely explained by detection of circulating basophils at baseline that could be activated by MOv18 IgE. The BAT assay was used to avoid enrolling further patients with reactive basophils. The safety profile is tolerable and maximum tolerated dose has not been reached, with evidence of anti-tumour activity observed in a patient with ovarian cancer. These results demonstrate the potential of IgE therapy for cancer.

All monoclonal antibodies in clinical use for the treatment of cancer belong to the IgG class, the most prevalent immunoglobulin in human blood[1]. Circulating levels of IgE are much lower than IgG, and IgE has evolved to provide immune surveillance in tissues[2–4]. The mechanism of action of therapeutic antibodies is mediated in part by their engagement with cognate Fc receptors on specific immune effector cell populations[5–7]. The structure of the Fc domain of IgE differs from that of IgG, and it binds with very high affinity to FcεRI receptors, expressed on cells including monocytes/macrophages, mast cells, basophils, and dendritic cells, resulting in long retention of antibody

by these effector cells in the absence of immune complex formation[8] (Fig. 1a). By contrast, the affinity of IgG for its Fc receptors, present on effector populations such as natural killer (NK) cells and monocytes/macrophages, is 100–10,000-fold lower[9]. IgE antibody drugs can mediate a more potent immune response to cancer cells than IgGs because of this higher affinity for Fc receptors, which in turn are expressed on an effector cell population distinct from those for IgG.

Preclinical comparison of anti-tumour activity using an IgE or an IgG antibody, each specific for the same target antigen, has demonstrated that the efficacy of IgE is indeed superior, both in human tumour xenograft-bearing mouse models[10–13] and in an immunocompetent rat model of metastatic cancer[14,15]. IgE induces antibody-dependent cell-mediated killing of cancer cells by both cytotoxic and phagocytic mechanisms[10–12,16] mediated by monocytes and macrophages. These effects are coordinated by secreted mediators, such as TNFα, MCP-1 and IL-10 in tumours (Fig. 1b)[14,17]. We hypothesise that, in line with these studies using animal models, the higher affinity of IgE binding to Fc receptors on distinct effector-cell populations may result in superior clinical efficacy of IgE therapies compared to IgG. The first step in testing this hypothesis is establishing whether IgE drugs can be safely administered to humans.

The human folate receptor-alpha (FRα) was chosen as the target for clinical testing of IgE therapy because of its presence on the cell membrane (Fig. 2a) of a range of tumour types, and very limited expression on normal tissues[18,19]. IgG-based therapeutics specific for FRα have shown evidence of anti-tumour activity in trials[20–24].

We have designed and conducted a Phase I trial of MOv18 IgE, a chimeric monoclonal IgE antibody specific for FRα, in patients with tumours expressing this antigen. Procedures prior to intravenous administration of MOv18 IgE were included to mitigate the possibility of unwanted IgE-mediated effects. First, skin prick testing with a solution of MOv18 IgE antibody was undertaken, with a rationale of detecting putative soluble factors, possibly originating from the target tumour, that might cross-link the antibody once bound to Fc receptors. This could result in mast cell and basophil degranulation and possible anaphylaxis. If present these factors would be expected to produce a wheal and flare reaction on skin prick testing, analogous to those seen in classical cutaneous allergy testing performed with solutions of antigens[25]. Secondly, pre-dose basophil activation tests (BAT)

were performed as previously described[26] on fresh whole blood from each patient using MOv18 IgE. This test was designed to model ex vivo the ability of MOv18 IgE to activate circulating cells, with the expectation that the BAT would function as an assay predictive of anaphylaxis. The safety profile of MOv18 IgE is tolerable, the most common toxicities being cutaneous, and there is evidence of anti-tumour activity.

## Results
### Dose escalation
Patients with any solid tumour expressing FRα using our immunohistochemical assay (Fig. 2a, b) were eligible, although all patients recruited to this trial had ovarian epithelial cancer (n = 21), tubal carcinoma (n = 3) or endometrial cancer (n = 2). A low prevalence of FRα expression was observed in other tumour types screened using our validated assay. 24 patients (median age 61) were treated with up to six weekly MOv18 IgE infusions (Fig. 2c) after intradermal or skin prick testing (Fig. 2d; see Methods) at escalating doses ranging from the starting dose of 70 µg to 12 mg (Fig. 2e). Patients potentially benefitting (stable disease or response), and without intolerable toxicity, could continue with three subsequent two-weekly maintenance doses. All patients had received prior systemic therapy, and in many cases were multiply pre-treated. The first patient developed symptomatic brain metastases after one 70 µg dose, and so was withdrawn and replaced. A dose-limiting toxicity (DLT) of grade 3 urticaria occurred in the second (250 µg) cohort, which was expanded. No further DLT was seen at this dose, but 5 patients in this cohort demonstrated a reaction to intradermal testing with MOv18 IgE prior to their first or second dose and were withdrawn. Intradermal testing was replaced by the skin prick test for subsequent recruitment, and 3 further patients safely received multiple 250 µg doses. A dose-limiting toxicity of anaphylaxis occurred in the third (500 µg) cohort, which was expanded to 6 patients without further DLT. Three patients were then treated at each of 750 µg, 1.5 mg, and 3 mg dose levels. Subsequently, based on a predictable safety profile in earlier cohorts, the protocol was amended to allow intra-patient dose escalation, and a further patient was treated at 6 mg and then 12 mg (cohorts 7$_{(i)}$ and 7$_{(ii)}$, respectively). No further DLTs occurred and a maximum tolerated dose has not been defined.

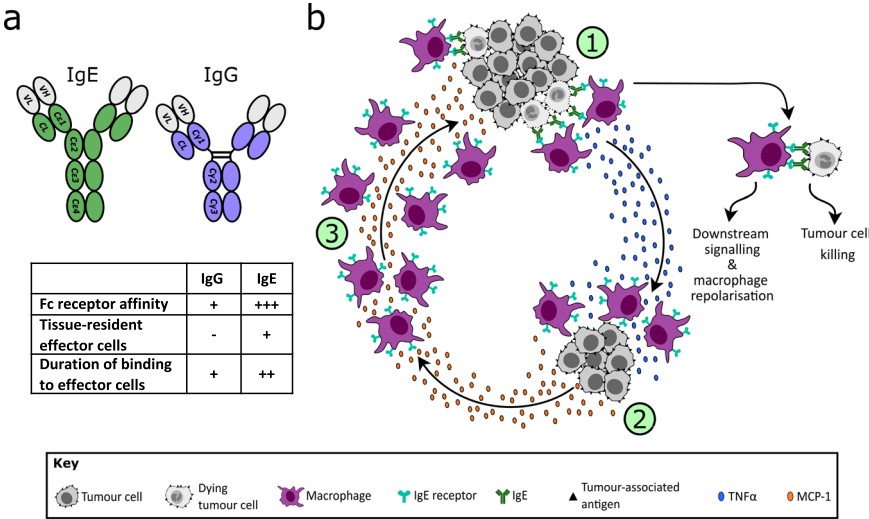

**Fig. 1 | Mechanism of cytotoxicity mediated by MOv18 IgE. a** The structurally distinct Fc domains of IgE and IgG underpin differences in their biological characteristics, supporting exploration of IgE therapies as potentially superior to IgG [refs. 8,9]. **b** (1) IgE bound to Fc receptors on macrophages is crosslinked by antigen expressed on tumour cells, leading to target cell cytotoxicity, release of proinflammatory mediators (e.g. TNFα) and macrophage repolarization. (2) TNFα upregulation in turn triggers MCP-1 production by monocytes and tumour cells, followed by (3) recruitment of further macrophages, mediated by MCP-1 [refs. 10,12,14].

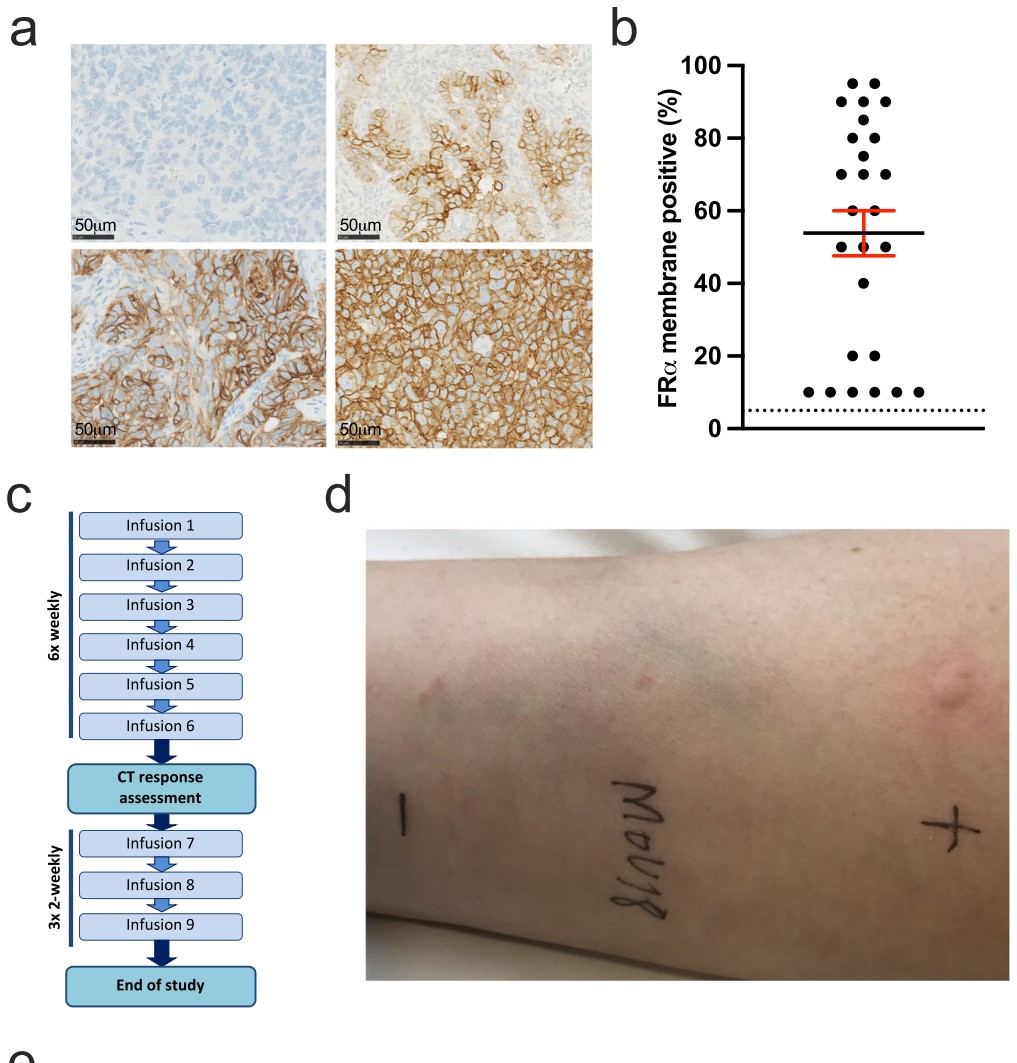

**Fig. 2 | Patient selection and study treatment. a** Representative immunohistochemistry of paraffin-embedded ovarian cancers showing a range of FRα membrane expression on 10 to 95% of tumour cells. **b** FRα membrane expression for all enroled patients (mean ± SEM; dotted line represents the 5% expression threshold required for trial eligibility; $n = 26$). **c** The trial dosing schedule consisted of 6 weekly doses of MOv18 IgE, before first on-treatment tumour assessment. **d** Skin prick testing comprised histamine positive control [+], saline negative control [−] and MOv18 IgE solution. The result of this typical skin prick test was negative. **e** MOv18 IgE dose escalation, and number of patients treated in each cohort ( *cohort 2 included 2 patients not evaluable for safety who did not receive intravenous dosing because of positive intradermal tests (see text); †a DLT occurred in these cohorts; ‡intra-patient dose escalation). Source data are provided as a Source Data file.

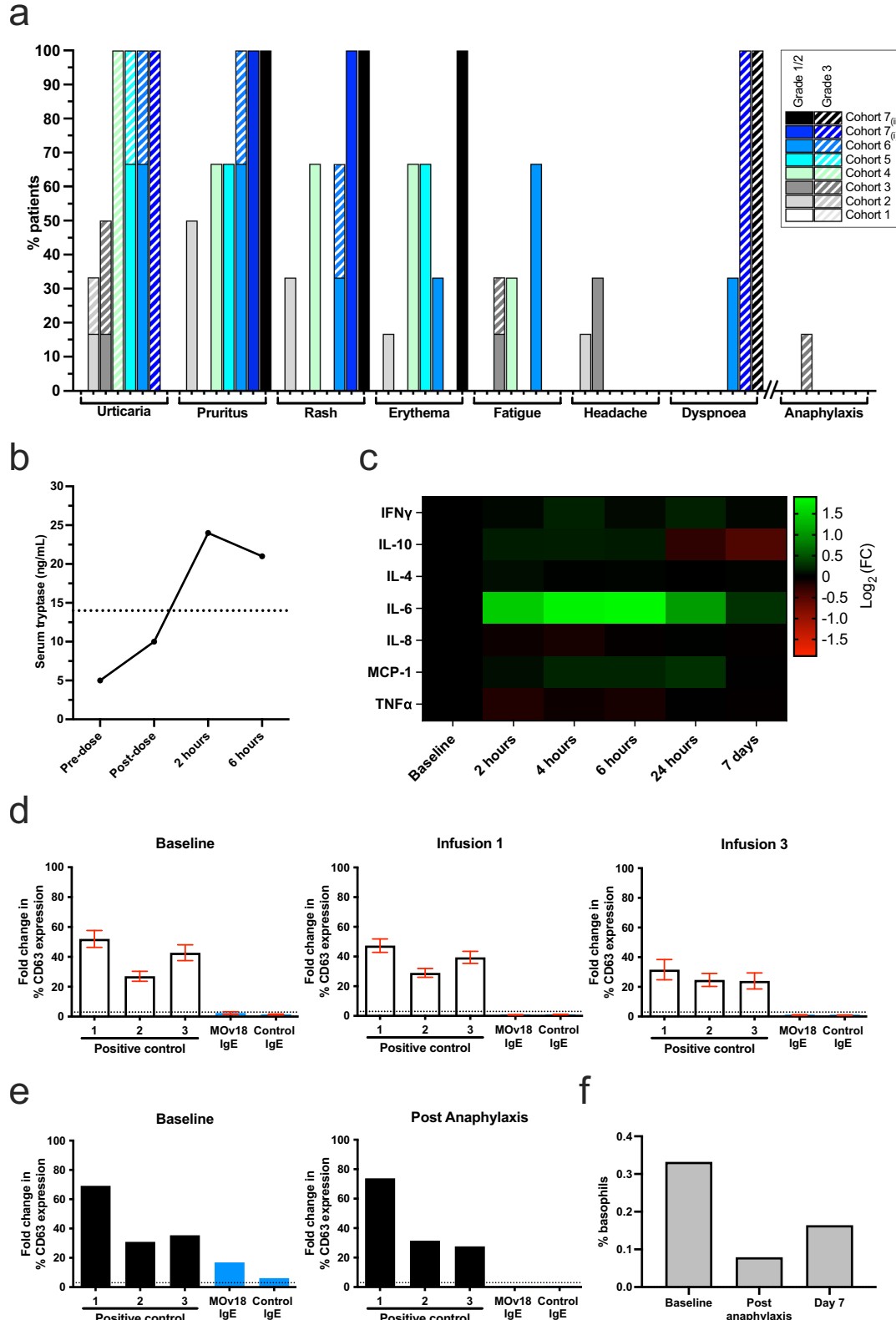

## Safety

MOv18 IgE was generally well tolerated with the great majority of adverse events being low grade (≤2, NCI Common Terminology Criteria for Adverse Events version 4.0). The most common events were localised cutaneous toxicities including urticaria, pruritus and rash (Fig. 3a); cutaneous events appeared to be dose-related. Urticaria was often associated with pruritus, less prominent with repeat

dosing, and in all but one case (described below) not associated with any systemic signs, symptoms or elevation of serum tryptase. Urticaria always resolved within hours of dosing, either spontaneously or with the administration of systemic steroids and antihistamines. Secondary prophylaxis with these supportive medications was allowed in patients experiencing urticaria with a previous dose.

**Fig. 3 | Treatment-related adverse events, and basophil activation test (BAT) as a predictor of anaphylaxis. a** Adverse events (AEs) experienced by ≥10% of patients (as well as anaphylaxis in 1/26 (4%)) are grouped according to severity and by dose cohort (1–7(ii) from left to right for each AE). No related AEs occurred at the lowest dose (cohort 1), and there was no grade 4 or 5 toxicity at any dose. Acute hypotension responding promptly to intramuscular adrenaline was seen in 1 patient treated at 500 µg (cohort 3). **b** Serum tryptase was elevated in this patient at 2 and 6 h following the infusion, supporting a diagnosis of drug-related anaphylaxis to explain this adverse event (dotted line represents upper limit of normal for serum tryptase [14 ng/ml]). **c** Fold change (Log$_2$) in median circulating cytokine levels post-dose (2, 6, 24 h and 7 days) relative to normalised baseline values for each patient ($n = 26$). **d** Marked upregulation of the basophil activation marker CD63 in response to ex vivo stimulation of patient blood using positive controls (1 = anti-FcεRI, 2 = fMLP, 3 = anti-IgE), but not to MOv18 or control IgEs. This BAT result ($n = 26$) remained negative when performed following the 1st ($n = 24$) and 3rd ($n = 20$) MOv18 IgE doses (mean ± SEM) for all trial subjects excluding the single patient who experienced anaphylaxis. Negative control IgE was not included in all assays. **e** By contrast, the baseline BAT was positive for the patient experiencing anaphylaxis at the time of their subsequent MOv18 IgE infusion ($n = 1$). Unlike all other patients, MOv18 IgE antibody produced a 17-fold increase in basophil activation in this individual's blood at baseline (left panel, blue bars). The BAT for this patient became negative after the episode of anaphylaxis (right panel), likely as a result of basophil depletion. **f** Peripheral basophil counts fell immediately following MOv18 IgE infusion and anaphylaxis in this patient, but not in other trial subjects after their MOv18 IgE dosing (Supplementary Fig. 1), recovering by day 7 ($n = 1$). Source data are provided as a Source Data file.

One patient treated in the 500 µg cohort experienced an episode of transient hypotension during their first infusion, a few minutes following the onset of urticaria. Treatment was discontinued and the hypotension responded immediately to adrenaline and the administration of intravenous hydrocortisone, antihistamine and saline. The diagnosis of grade 3 anaphylaxis was confirmed by the detection of elevated serum tryptase following this episode (Fig. 3b), which was not seen after any of the episodes of isolated urticaria observed with multiple MOv18 IgE doses in other patients (not shown). Similar treatment-related changes in circulating cytokine concentrations were detected in all patients (median values for the trial population shown in Fig. 3c). Although the basophil activation test (BAT), using MOv18 IgE stimulation, had been negative for all other patients treated in the trial both before and after treatment (Fig. 3d), this patient's BAT had been positive during screening prior to trial treatment (Fig. 3e). This systemic toxicity was not evident in any other patient, and because the mechanism was not considered to be related to dose, the protocol was amended with ethics committee approval to allow continued dose escalation, but only in patients with a negative baseline BAT. We hypothesised that a positive BAT at baseline predicts the possibility of systemic anaphylaxis developing as a result of MOv18 IgE therapy. The dose was subsequently safely escalated in all the remaining trial patients, each with a negative BAT, to 12 mg without repetition of anaphylaxis. No other DLTs were observed and MTD was not reached.

### Pharmacodynamics

Skin prick testing was routinely performed before each intravenous dose. In general, the reaction to MOv18 IgE in this test was indistinguishable from the negative diluent control, although a wheal reaction of up to 3 mm was also considered negative (Fig. 2d). A positive cutaneous reaction to skin prick testing with MOv18 IgE was not observed in any of the patients in the study, including the patient who developed anaphylaxis following intravenous dosing. By contrast the performance of the BAT did distinguish the single patient who experienced anaphylaxis, exhibiting a positive BAT at baseline which became negative immediately following the event (Fig. 3e), then positive again by day 7 of follow-up (not shown), consistent with recovery of a functional circulating basophil population (Fig. 3f). Basophil depletion was not observed following infusion of MOv18 IgE in any other patient (Supplementary Fig. 1).

Free FRα protein was detected at one or more time points (baseline, prior to infusion 4 and at 28-day follow up) in the serum of eight patients, although in the remainder it was undetectable throughout (Supplementary Fig. 2a). Circulating anti-FRα antibodies were detected in only three patients, at the 28-day follow up, and in the remainder they were undetectable throughout (Supplementary Fig. 2b). One patient had detectable concentrations of both serum FRα and anti-FRα auto-antibodies at a single timepoint.

Analysis of circulating cytokine concentrations in all the treated patients revealed an increase in the mean fold-change serum concentration of IL-6 from baseline, which peaked 4–6 h after dosing and was detectable for up to 7 days (Fig. 3c). Mean fold-change serum concentrations of IFNγ and MCP-1 were similarly increased modestly within 24 h of dosing, while those of IL-10 showed a modest increase within 6 h, followed by a fall between 24 h and 7 days following dosing. Fold-change concentrations of circulating levels of TNFα and IL-8 were marginally decreased following dosing. No changes in IL-4 were observed (Fig. 3c).

Alpha-GAL IgE antibodies[27,28] were measured in patient sera before, during and after MOv18 IgE treatment. At baseline, no patient had alpha-GAL antibody titres >0.1 kU$_A$/L (threshold of positivity according to reference laboratory). Three patients developed anti-alpha-GAL antibodies (>0.1 kU$_A$/L, patient values ranged from 0 to 0.99 kU$_A$/L) but these were not associated with any clinical signs of anaphylaxis and were no longer measurable at the 28-day follow-up (Supplementary Fig. 3).

### Anti-drug antibody responses

Samples for the ADA assay were collected from 26 patients. ADA detection was definitively confirmed, at 1 or 2 time points, in only 3 patients (Supplementary Fig. 4). There was no correlation between ADA titre and toxicity or clinical benefit.

### Pharmacokinetics

Serum MOv18 IgE clearance profiles in each individual patient are shown in Fig. 4a. Systemic exposure, as measured by AUC$_{(0-24h)}$ and $C_{max}$, was observed to increase in a dose-proportional manner (Fig. 4a, b). The mean terminal elimination half-life was 9.4 h (Fig. 4c), mean total body clearance was 2.81 L/h, and these parameters were consistent across the range of doses studied. In six of 24 patients who received at least one MOv18 IgE infusion, $C_{max}$ exceeded the baseline endogenous IgE level in that patient (endogenous IgE levels are shown in Supplementary Fig. 5).

### Anti-tumour activity

Twenty of 24 patients, who had at least one on-treatment CT scan, were evaluable for efficacy (Fig. 5a). Six patients continued dosing into the maintenance period, with a best response of stable disease according to RECIST (Response Evaluation Criteria in Solid Tumours) criteria[29]. One patient with platinum-resistant mixed high grade serous and endometrioid ovarian carcinoma treated at 700 µg/week exhibited a 52% reduction of the serum CA125 tumour marker concentration in cycles 1 and 2 (Fig. 5b). This marker reduction was not sustained beyond 28 days, therefore not meeting formal Gynaecologic Cancer Intergroup criteria for response[30]. This patient's on-treatment scan after 6 weeks revealed resolution of ascites and short-lived shrinkage of peritoneal tumour deposits (not amounting to RECIST partial response; Fig. 5c). The scan and tumour marker at 12 weeks demonstrated disease progression, following per protocol reduction of dosing frequency to two-weekly. This evidence of anti-tumour activity was seen in a patient who had derived little benefit from conventional chemotherapy. Their previous treatment course was notable for

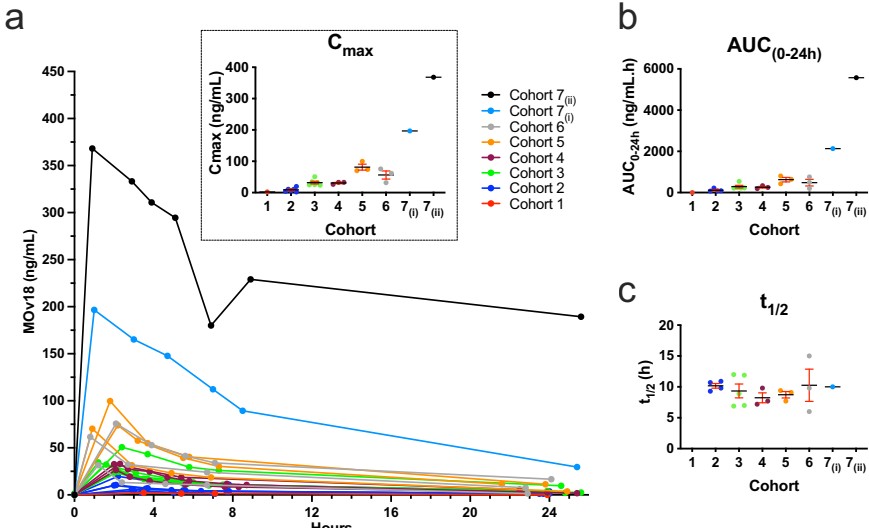

**Fig. 4 | Plasma pharmacokinetics of MOv18 IgE. a** Linear plot of plasma drug concentrations for evaluable patients ($n = 23$) showing dose-proportional $C_{max}$ (inset; mean ± SEM; $n = 23$), which exceeded baseline endogenous IgE levels in 25% of patients. **b** Area under the curve (AUC$_{(0-24h)}$; mean ± SEM; $n = 20$) also increased with dose. **c** Half-life (mean ± SEM; $n = 19$) was consistent in the range of doses tested, with a mean $t_{1/2} = 9.4$ h across all cohorts. Some measurements were below level of quantification for MOv18 IgE. Source data are provided as a Source Data file.

relapse within five months of adjuvant carboplatin and paclitaxel, followed by six cycles of weekly paclitaxel without response. At the time of trial enrolment this patient was aged 62 and had a performance status of zero. Membrane FRα expression was observed on 10% of their tumour cells (range for recruited patients 10–95%).

## Discussion

In this clinical trial of an IgE antibody for the treatment of cancer, a manageable safety profile, distinct from that of IgG drugs, was observed and preliminary evidence of efficacy demonstrated. Both anti-tumour activity and adverse events occurred at doses very much lower than typically observed for IgG antibodies. This reflects fundamental differences in Fc-receptor affinity and effector cell biology[3,31], including the absence of an inhibitory IgE receptor, as well as lower circulating serum concentrations of endogenous IgE in humans. Serum concentrations of MOv18 IgE achieved during treatment exceeded those of endogenous IgE for several patients enrolled in this study. The plasma half-life of MOv18 IgE was substantially shorter than observed with IgG therapeutic antibodies, as might be expected given the rapid clearance of IgE, compared with antibodies of other isotypes, from the circulation of healthy subjects, regardless of atopic status[32]. This likely reflects avid binding of the IgE molecule to Fcε-receptors expressed on effector cells such as basophils and monocytes[10,12,26]. These cells readily traffic to tissue extravascular space or are already resident there.

Our finding of elevated fold increases in serum concentrations of IL-6, IFNγ, MCP-1 and IL-10 in treated patients is consistent with the mechanism of IgE-induced signalling by tumouricidal effector cells, such as monocytes and macrophages, described in animal surrogates (Fig. 1)[14,17,33]. Although the mean serum IL-6 concentration was increased in the 24-h period following MOv18 IgE treatment, we did not observe any of the IL-6-mediated toxicity associated with uncontrolled immune activation complicating some advanced therapies. The modest decrease in circulating TNFα in our patients may not fully reflect the changes in this cytokine in the tumour microenvironment. Insufficient pre- and post-treatment tumour biopsies were available for formal immunophenotyping in this trial. Serum IL-4, a cytokine characteristically implicated in allergy, did not change with MOv18 IgE treatment in either animal models or patients. This confirms the impression that the anti-tumour mechanism of IgE in this trial likely reflects its ability to activate cell-mediated immunity against tumour cells and parasitic pathogens, rather than involvement in a conventional Type 1 allergic reaction[4,17,34–36].

In this trial we performed a basophil activation test (BAT) ex vivo on whole unfractionated blood from each patient just prior to dosing, with a rationale of addressing concerns that MOv18 IgE bound to endogenous Fcε receptors might be cross-linked by endogenous antibodies or other circulating factors. The BAT results remained negative in all patients with the exception of one who developed anaphylaxis on their first exposure to MOv18 IgE. The episode resolved rapidly and did not recur after withdrawal of drug. This patient demonstrated a positive BAT at baseline, prior to any exposure to MOv18 IgE. We hypothesise that, in this single patient, both anaphylaxis and the positive BAT at baseline reflected the presence of an endogenous circulating factor able to crosslink the MOv18 IgE antibodies bound to receptors on the surface of effector cells. One such candidate may be autoantibodies to the oligosaccharide galactose-α−1,3-galactose (alpha-GAL): subjects with an endogenous IgE response against this post-translational modification on a therapeutic antibody have been reported to develop Type 1 hypersensitivity reactions during treatment with some monoclonal antibodies[27]. Development of anti-alpha-GAL IgE antibodies following treatment with MOv18 IgE was detected in three patients in our trial but was not associated with any clinical sequelae. Furthermore, seropositivity to alpha-GAL was not detected, before or after treatment with MOv18 IgE, in the patient who experienced anaphylaxis (Supplementary Fig. 3). We also considered the possibility that release of FRα from tumour cells might have cross-linked MOv18 IgE bound to effector cells such as mast cells and basophils. However, serum concentrations of FRα and anti-FRα auto-antibodies were undetectable in this patient at all timepoints. Our previous ex vivo studies suggest that even large numbers of circulating tumour cells expressing FRα are not sufficient to cross-link IgE[37].

Following this single case of anaphylaxis, the trial protocol was amended to exclude any further patients found to have a positive baseline BAT, and no further anaphylaxis was observed. In a separate observational study of blood samples drawn from a series of patients with ovarian cancer, we observed a positive BAT using MOv18 IgE in only one of 42 individuals[26]. The prevalence of positive baseline BAT

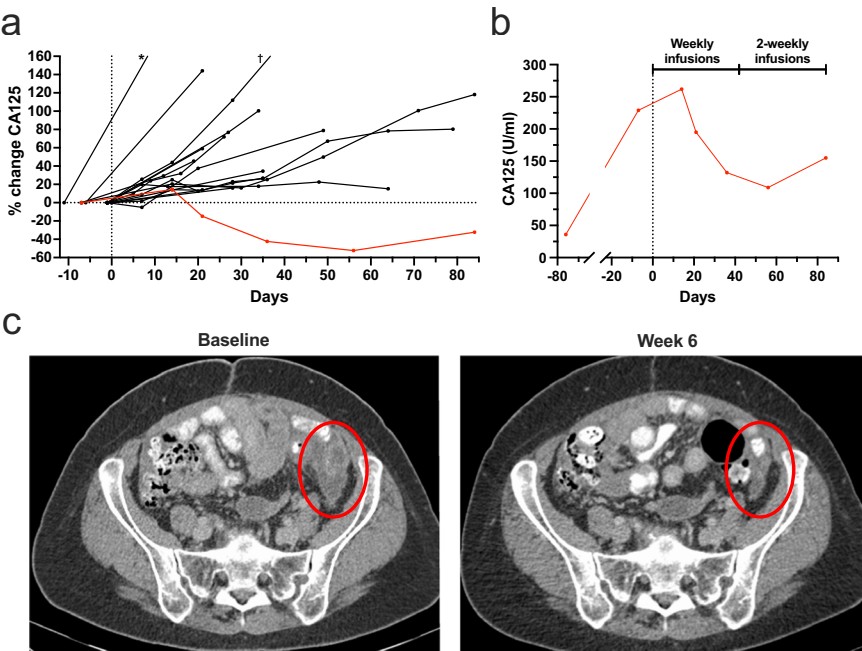

**Fig. 5 | Anti-tumour activity. a** Percentage change in CA125 circulating tumour marker for evaluable patients (*n* = 17; vertical dotted line indicates initiation of dosing; red line indicates a patient with marked CA125 reduction and tumour volume reduction; * and † = tumour marker progression >200%). **b** Marked but short-lived CA125 reduction following MOv18 IgE dosing in a patient with progression prior to treatment. CA125 relapse coincided with change from weekly to alternate-weekly dosing after 6 weeks. **c** Pelvic CT images for this patient at baseline and week 6 of treatment showing reduction of peritoneal tumour deposit (red oval). Source data are provided as a Source Data file.

therefore appears low in this population of patients with cancer, but screening for baseline BAT positivity may be useful in future to exclude these infrequent patients from IgE therapy and the associated risk of anaphylaxis.

The most common adverse events associated with MOv18 IgE were cutaneous, with some patients experiencing a diffuse patchy macular erythematous rash, with or without pruritus, but with urticaria the most frequent toxicity. Although the extent of this rash varied greatly between patients, it was tolerable and transient. Urticaria usually recurred with subsequent doses, but in general it was more prominent with first dose administration and became less obvious with subsequent doses. It is possible this could have been a result of the use of secondary prophylaxis with steroid and antihistamine. Urticaria was associated with symptoms or signs of anaphylaxis only in the single patient referred to above with a positive BAT at baseline. No elevation of the serum tryptase concentration, a diagnostic requirement in cases of anaphylaxis[38], was observed in any of the remaining patients who experienced urticaria. Cutaneous reactions were not seen in animal models of IgE therapy[15,39], and the mechanism accounting for the common occurrence of urticaria, in the absence of anaphylaxis, in humans receiving MOv18 IgE is currently unclear. Our own data, consistent with the literature[40,41], suggest that FRα is not expressed in skin (Stavraka, personal communication), and although distribution to the skin of healthy subjects was claimed in a clinical study of a fluorescently labelled anti-FRα agent, this was attributed to drug aggregation[42]. MOv18 IgE does not bind to human skin ex vivo, and histological examination of biopsied urticarial skin from a patient in this trial did not demonstrate prominent infiltration by basophils (C Stavraka, personal communication).

FRα was chosen as the target for testing IgE therapy in a clinical trial. This tumour-associated antigen is present at very low levels in normal tissues[43], but is expressed in a range of solid tumours[18]. Several IgG-based antibody therapies targeted to FRα have been tested clinically[20,22,23,44], with the recent approval of mirvetuximab soravtansine further validating this target[24]. Our pre-clinical evidence supporting superior anti-tumour activity of FRα-targeting IgE suggests a route to improvement of earlier clinical results with anti-FRα IgG. Tumour types most commonly expressing FRα with the IHC used in this trial were epithelial ovarian/tubal and endometrial cancers. We observed preliminary evidence of MOv18 IgE anti-tumour activity in high grade serous ovarian carcinoma, with a fall in CA125 and tumour shrinkage in a patient with chemotherapy-resistant disease. If this is confirmed in subsequent clinical development, this antibody may join the growing immunotherapy armamentarium in FRα-positive cancers[45]. More broadly, these results offer the opportunity to generate a new class of IgE antibody cancer therapies specific for other antigens already validated as targets for IgG drugs.

## Methods
### Study design
This was an open-label, dose-escalating Phase I study of MOv18 IgE conducted at 4 sites in the United Kingdom. The study was undertaken under the sponsorship and management of the Cancer Research UK Centre for Drug Development, conducted in accordance with International Council for Harmonisation Good Clinical Practice guidelines and ethical principles of the Declaration of Helsinki, and approved by the UK Medicines and Healthcare products Regulatory Agency and National Health Service Health Research Authority (EudraCT number 2014-000070-19; ClinicalTrials.gov identifier NCT02546921, first registered 11 September 2015). Participants provided written informed consent prior to the conduct of any study procedures. Antibody was administered as an intravenous infusion weekly for a six-week treatment period. In the absence of disease progression or intolerable treatment-related toxicity, patients were offered the option to continue two-weekly dosing for a further maintenance period.

Screening tests at baseline included physical examination, computerised tomography (CT), standard tests of organ function, beta-tryptase, endogenous IgE level, tumour FRα expression, basophil activation test, and intradermal or skin-prick test. Safety assessments were repeated prior to each further dose throughout the study. All

patients receiving at least 1 intravenous dose of MOv18 IgE were evaluable for safety. Anti-tumour activity was assessed using standard tumour markers where relevant (for example, CA125 in ovarian cancer cases), and CT according to RECIST version 1.1 at intervals of 6 weeks. Laboratory endpoints were analysed using SSPS version 29 and presented with Graphpad Prism version 9.5.

## Patients

Recruitment and treatment took place between 23 February 2016 and 20 April 2021. Eligible patients had advanced or metastatic solid tumours not suitable for alternative standard treatment and were over 16 years of age. Immunohistochemical evidence of tumour FRα membrane expression was required (based on a criterion of ≥5% tumour cells with membrane positivity)[46]. Briefly, sections of formalin-fixed, paraffin embedded tissue were pre-treated using citrate-based heat-induced antigen retrieval (CC2) for 92 min, followed by staining with FRα antibody (NCL-L-FRα, Leica Novocastra) at a dilution of 1:500 using the Ventana Benchmark Ultra (Roche) automatic staining system. Positive (FRα expressing ovarian tumour tissue) and negative (normal ovarian tissue) controls were included. Of the 445 individuals assessed for tumour expression of FRα, 37% were deemed to be positive. Other inclusion criteria were: measurable or evaluable disease; resolution of any residual toxic effects, aside from sensory neuropathy, related to prior anti-cancer therapy to grade 1 or lower (as defined by the Common Terminology Criteria for Adverse Events (CTCAE), version 4.0); no chemotherapy, radiotherapy, biological therapy, or hormone for at least four weeks, or five half-lives, prior to receiving the study treatment; an Eastern Cooperative Oncology Group (ECOG) performance score of 0 or 1; adequate bone marrow, liver, and kidney function. Patients with treated and stable brain metastases were allowed. Exclusion criteria included: a history of congenital or acquired immunosuppression, including an ongoing requirement for systemic steroid therapy; high risk or uncontrolled asthma; a history of anaphylaxis; elevated serum tryptase at baseline; a history of laryngeal oedema, known HIV, hepatitis B or C infection; pregnancy or breast-feeding; elevated risk in the event of anaphylaxis because of heart failure, uncontrolled infection, vascular disease, previous cerebrovascular accident, extensive lung metastases or pleural effusion; concurrent medication with drugs likely to inhibit or augment the effects of adrenaline, including beta-blockers, tricyclic antidepressants and monoamine oxidase inhibitors.

## Drug administration

Drug was supplied (Cancer Research UK) and manufactured as single-use aseptic 1 mL fill in 2 mL glass vials containing a solution of MOv18 IgE at a concentration of 1 mg/mL at pH6.5 with 0.1 M sodium citrate, 30 g/L L-arginine, 50 g/L sucrose and 0.02% polysorbate 20 in water for injection. Stock was stored at 5 ± 3 °C, initially diluted to a concentration of 100 μg/mL in 0.9% sodium chloride, then infused over 90 min through a peripheral line in a total volume of 250 ml 0.9% saline. A shorter infusion time for subsequent doses of 60 min was allowed if no adverse reactions are encountered. Vital signs were monitored during the infusion and at regular intervals afterwards. Patients were hospitalised overnight following the first administration to facilitate safety monitoring procedures.

The protocol-defined treatment regimen was 6 doses of MOv18 IgE administered weekly, following which disease response was reassessed using CT. Patients tolerating treatment and appearing to benefit from treatment were offered the option of continuing into a maintenance phase of two-weekly dosing for a further 6 weeks (Fig. 2c).

## Dose escalation

The planned escalation in successive flat dosing cohorts was 70 μg, 250 μg, 500 μg, 700 μg, 1.5 mg, 3 mg, 6 mg and 12 mg total protein. An accelerated dose escalation scheme, starting with single-patient cohorts, was planned up to cohort 5 (1.5 mg)[47]. An additional 2–5 patients could be added to each cohort in the event of emergent toxicity. Thereafter, three patients were enrolled per cohort, with an additional three patients added to a cohort if needed for toxicity. The starting dose of 70 μg was selected to be lower than endogenous IgE, and was based on the relatively low physiological IgE serum levels and high affinity for receptors on effector cells. It also reflected partial occupancy of Fc receptors by endogenous IgE, meaning that anti-tumour IgE could engage and activate immune cells at concentrations much lower than the equivalent IgG. Each patient was assessed for safety and dose-limiting toxicities (DLTs). DLT was defined as any grade 3/4 non-haematological adverse event (with exceptions in the case of incomplete supportive medication for diarrhoea or vomiting, grade 3 fatigue, or biochemical abnormalities deemed clinically non-significant) occurring in the first 3 weeks of MOv18 IgE administration. Haematological DLTs were defined as grade 4 neutropenia or thrombocytopenia that was prolonged, or grade 3/4 neutropenia associated with fever or infection.

## Cutaneous testing and basophil activation test

Skin prick testing (SPT) with a solution of MOv18 IgE antibody was undertaken in patients prior to each intravenous administration. Positive histamine and negative saline controls were included, and the presence of a wheal reaction in response to the positive control was required for a test to be considered valid (Fig. 2d). Concomitant dosing with antihistamines was not allowed within 4 half-lives before SPT. For early patients recruited to the trial, intradermal testing was used instead of SPT [https://www.bsaci.org/Guidelines/SOPs], but subsequently SPT was preferred due to serial positive results with intradermal testing, which did not appear to discriminate those at risk.

Similarly, basophil activation tests (BAT) were performed as previously described[26] on fresh whole blood using aliquots of the trial supply of MOv18 IgE, and a laboratory standard solution. Briefly, basophils in unfractionated whole blood samples were incubated with positive controls anti-FcεRI, fMLP (each as part of Buhlmann FlowCAST kit FK-CCR-U), and anti-IgE (Dako Aligent A0094), or MOv18 and control IgE antibodies (non-FRα-reactive IgE antibodies) for 30 min at 37 °C. Staining antibodies (Buhlmann FlowCAST kit FK-CCR-U) were used for CCR3 (to identify basophils) and CD63 (as a marker of activation)[48]. Flow gating strategy is shown in Supplementary Fig. 6. Data were analysed as the fold change in percentage CD63 expression relative to the patient's background. BAT was performed at baseline and after administration of the first and third intravenous doses to each patient, as well as immediately following any systemic adverse event.

## Other pharmacodynamic assays

No features of anaphylaxis or other manifestation of allergic toxicity were observed in preclinical animal models, but in addition to the risk mitigation steps described above, serial measurements of serum tryptase were included in this trial. Tryptase is released by degranulation of activated effector cells, and elevation is characteristic of anaphylaxis[38]. The protocol mandated tryptase measurement at baseline, at the end of each infusion, and following any infusion-related event, primarily to distinguish cytokine release from anaphylaxis. It was also measured following the onset of urticaria, which can occur as a feature of anaphylaxis[49].

Patient specimens were also collected for pharmacodynamic analysis. Serum samples were collected pre- and post-dose for multiplex analysis of a cytokine panel (Randox) relevant to IgE biology[15,17] and to the proposed mechanism of anti-tumour activity of MOv18 IgE[14,50]. Where possible, serial tumour biopsies were collected for analysis of changes in the tumour microenvironment.

Anti-drug antibody (ADA) samples were drawn at baseline and 2 and 5 weeks following initiation of weekly dosing. Additional samples were collected 28 days after the last dose was administered and following any suspected infusion reaction. Serum samples were stored frozen at −80 °C prior to analysis using a bridging ELISA assay with MOV18 IgE to capture, and biotinylated MOv18 IgE followed by streptavidin-HRP (BD 554066) to detect the presence of ADAs. A confirmatory assay was performed on all positive samples, pre-incubating with drug to distinguish specific responses from false positives.

Further blood samples were drawn at baseline, 3 weeks following initiation of weekly dosing and 28 days after the last dose, as well as following any observed adverse events. Serum was separated and stored at −80°C prior to evaluation of circulating soluble FRα and anti-FRα antibodies by ELISA, as previously described[37]. Briefly, FRα in patient sera was captured by mouse anti-human FRα IgG1 (R&D Systems MAB5646), and detected by biotinylated polyclonal goat anti-human FRα antibody (R&D Systems BAF5646) followed by streptavidin-HRP (Pierce 21130). Similarly, anti-FRα antibodies were captured by recombinant FRα (BioTechne 5646-FR) and detected by anti-IgG1-HRP antibody (Jackson Immuno Research 109-036-098). Levels were quantified by recombinant FRα (BioTechne 5646-FR) and anti-FRα IgG1 (in house) standard curves, with lower levels of quantification (LLOQ) of 6.25 ng/ml and 3.125 ng/ml in serum, respectively.

## Pharmacokinetic sampling and assays

Pharmacokinetic blood samples were drawn immediately pre-dose and at 0.5, 2, 4, 6, and 24 h, then 7 days, following the first dose, and 28 and 70 days after the final dose. Serum was separated from blood and stored at −70°C prior to analysis for MOv18 IgE concentrations by indirect ELISA, using a fully validated assay. Briefly, MOv18 IgE antibodies were captured by recombinant FRα (BioTechne 5646-FR) and detected by an anti-IgE-HRP detection antibody (Sigma A9667). Levels were quantified by a MOv18 IgE standard curve with a limit of quantification of 2.5 ng/ml and a dynamic range of 2.5–200 ng/mL in serum. The following pharmacokinetic parameters were evaluated: terminal half-life calculated from the terminal slope of the log concentration–time curve ($t_{1/2}$), maximum concentration ($C_{max}$), time of peak serum concentration ($T_{max}$), and area under the serum concentration-time curve (AUC).

## Reporting summary

Further information on research design is available in the Nature Portfolio Reporting Summary linked to this article.

# Data availability

The summary study data are available within this Article, Supplementary Information and Source Data file. The full clinical and laboratory data set that support the findings of this trial are held by Cancer Research Horizons, a wholly owned subsidiary of the trial sponsor Cancer Research UK. Tabulated de-identified patient data (including demographics and adverse event information) will be shared on request from the date of publication and will be retained for 15 years. Access requests are subject to review and approval by Epsilogen (info@epsilogen.com). The study protocol is also available on request from the sponsor (horizons@cancer.org.uk). Source data are provided with this paper.

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

## Acknowledgements

The trial was funded and sponsored by Cancer Research UK, who col-laborated with the academic authors in designing the trial, collated the clinical data, and reviewed the manuscript. The authors acknowledge additional financial support from the UK Department of Health and Cancer Research UK via Experimental Cancer Medicine Centre and NIHR Biomedical Research Centre grants to King's Health Partners/Guy's & St Thomas' NHS Foundation Trust (JS), the University of Cambridge/Cam-bridge University Hospitals NHS Foundation Trust (BB), University Col-lege London/UCL Hospital NHS Trust (RM), and Institute of Cancer Research/Royal Marsden Hospital (UB). We acknowledge the Immune Monitoring Core Facility team at Guy's and St Thomas' NHS Foundation Trust (IS-BRC-1215-20006) for flow cytometry facilities, and the Cancer Research UK King's Health Partners Centre at King's College London (C604/A25135). The authors would like to thank the patients, their families, the clinical research teams, as well as Sharmistha Ghosh, Amy Pope and Natalie Woodman for their contribution to this study.

## Author contributions

S.K., H.G., M.F., S.Can., D.J. and J.S. developed MOv18 IgE. J.S. (chief clinical investigator), S.K. (chief scientific investigator), C.C., C.B., P.J. and S.M. wrote the trial protocol. S.Car. and C.Sel. manufactured MOv18 IgE. J.S., S.K., H.B. and J.C. wrote a first draft of the manuscript, and all authors reviewed and approved the final submission. J.S., B.B., A.M., U.B., R.K., R.M., G.N., V.K., I.F., G.D. and C.Sta. recruited patients, col-lected clinical data and provided clinical care. C.C. and S.T. provided clinical allergy advice. S.M., P.J. and C.B. managed the trial and data-base. J.C., G.P., M.N., K.I., A.K., C.Sta. and H.B. developed and performed pharmacodynamic assays. G.V. developed and delivered pharmacoki-netic assays and data. S.P. and C.G. supervised tissue collection and the FRα immunohistochemical assay.

## Competing interests

J.S. and S.K. are co-founders of Epsilogen Ltd. H.B. is presently employed, and J.C. formerly employed, through a fund from Epsilogen Ltd. S.K., H.B., H.G., D.J, G.P. and J.S. hold patents on anti-tumour IgE antibodies. The remaining authors declare no competing interests.

## Additional information

[1]School of Cancer and Pharmaceutical Sciences, King's College London, London, UK. [2]Cancer Centre, Guy's and St Thomas' NHS Foundation Trust, London, UK. [3]Cambridge University Hospitals NHS Foundation Trust, and Cancer Research UK Cambridge Centre, University of Cambridge, Cambridge, UK. [4]Institute of Cancer Research and Royal Marsden Hospital NHS Foundation Trust, Sutton, UK. [5]University College London, London, UK. [6]Newcastle University Centre for Cancer, Newcastle upon Tyne, UK. [7]King's Centre for Lung Health, School of Immunology and Microbial Sciences, King's College London, London, UK. [8]ANP2, Department of Advanced Diagnostics, Fondazione IRCCS, Istituto Nazionale dei Tumori, Milan, Italy. [9]Fondazione IRCCS, Istituto Nazionale dei Tumori, Milan, Italy. [10]Centre for Drug Development, Cancer Research UK, London, UK. [11]Barton Oncology Ltd, Hertfordshire, UK. [12]Randall Centre for Cell and Molecular Biophysics, School of Basic and Medical Biosciences, King's College London, London, UK. [13]St. John's Institute of Dermatology, School of Basic and Medical Biosciences, King's College London, London, UK. [14]King's Health Partners Cancer Biobank, Guy's and St Thomas' NHS Foundation Trust, London, UK. [15]Breast Cancer Now Research Unit, School of Cancer and Pharmaceutical Sciences, King's College London, London, UK. [16]Present address: UCB Pharma Ltd., Slough, UK. ✉e-mail: james.spicer@kcl.ac.uk

