## [Peer Review File · Nature Communications]

Safety and anti-tumour activity of the first-in-class IgE antibody MOv18 in patients with advanced solid tumours: a phase 1 trialREVIEWER COMMENTS

Reviewer #1 (Remarks to the Author): with expertise in cancer clinical trials, folate receptor targeting

The manuscript presented the first clinical data that demonstrated the safety and effectiveness of an ADC using an IgE antibody for the treatment of ovarian cancer. The authors targeted the folate receptor alpha (FR α) for the clinical study and tested the IgE antibody candidate MOv18.

The safety profile was quite acceptable. Basophil activation and signs of systemic toxicity were observed in only one out of the 26 patient participants and that patient showed baseline basophil activation prior to the treatment. In general, the ADC induced only low-grade, manageable adverse effects among the patients, and produced an elevation of serum IL-6, as well as MCP-1 and IFN γ levels post treatment.

On the aspect of anti-tumor activity, MOv18 treatment led to a marked reduction of serum CA125 in cycle 1 and 2, as well as a reduction of tumor mass in one of the 24 patients, though the marker reduction was not maintained beyond 28 days. The study offered insights into the potential of IgE class antibodies as an alternative to the existing IgG class antibodies in the treatment of solid tumors. However, there are several points that may need to be clarified (listed below).

1. As indicated by the experimental design in Figure 2c, each patient received multiple infusions in the span of 10 weeks, including 6 weekly infusions and 2 biweekly infusions. However, the data that demonstrate the effect of Mov18 treatment in serum cytokine and tryptase levels are mostly showing only one cycle. Did the consecutive infusion cycles have any cumulative impact on the serum cytokines? Did the weekly and biweekly infusion cycles influence the serum cytokine levels differently? Did the adverse effects show any change in the subsequent infusion cycles? The authors may need to include a more thorough data/discussion that better explain the physiological impact of the treatment over the multiple infusion cycles.

2. One out of all the participating patients showed marked reduction of the serum level of CA125 marker and reduction of the tumor mass. Are there any tumor or patient correlatives which explain the sensitivity to MOv18 therapy? Do we know the FR α expression? The authors should discuss the case in more details.

3. It might be informative to discuss more about how the performance of MOv18 compare to that of IgG class FR α antibodies in anti-tumor activities, and how the antibody may benefit the patients more over existing IgG class antibodies.

Reviewer #2 (Remarks to the Author): with expertise in IgE biology, clinical

This is an important and interesting contribution. The phase I trial of of an IgE to folate receptor-alpha showed anti-tumour activity of this antibody in a patient with ovarian cancer.

The results are novel, of clinical significance, scientifically important, and well presented.

Major issues

None

Minor issues

The authors describe potential advantages of therapeutic IgEs of IgGs, i.e. 1) very high affinity to Fc ϵ RI receptors, 2) expressed on monocytes/macrophages, mast cells, basophils and dendritic cells, 3) long retention of antibody by these effector cells, 4) absence of immune complex formation, 5) more potent immune response to cancer cells. They refer to Fig 1, which is nice but doesn't really show the how therapeutic IgE differ from IgGs. Consider adding a table that describes these differences (and similarities) and includes refs in their support.

The authors performed skin prick testing and basophil testing with MOv18 IgE antibody to check for interaction with soluble factors, possibly originating from the target tumour. Why did the authors not check for

the presence of targets of MOv18 IgE antibody in the blood of patients, e.g. by ELISA or WB?

The authors say that patients potentially benefitting could continue with subsequent two-weekly maintenance dosing. How was this assessed? What were the criteria for "potential benefit"?

The authors state that most common events were localised cutaneous toxicities including urticaria, pruritis and rash. All three terms are problematic. Urticaria is a disease (defined by the occurrence of itchy wheals - also called hives, angioedema, or both. Use of the word "wheals" or "hives" instead of "urticaria" is advised. Substitute wheal for urticaria throughout. Pruritis = pruritus. Rash is too vague, describe the specific skin reaction(s) that occurred.

The antibody was administered as an intravenous infusion. Please add volume and infusion duration.

Control IgE antibodies were used for basophil testing. Which ones?

What were the patients' total IgE levels? Were these associated with efficacy and safety?

Did any of the patients have anti-IgE Abs?

In line 370, the authors refer to urticaria: The most common toxicity of MOv18 IgE was urticaria. Although the extent of this rash varied greatly between patients, it was tolerable and transient. Please see my comment above. Urticaria is vague, and rash is even more vague. What signs and symptoms did patients develop? Itchy wheals? If yes, when, how many, where, duration? Angioedema? If yes, when, where, duration? Rash is too unspecific. Avoid the use of this term.

Only one patient had clinical benefit. How was this patient different than the others? What may be the explanation for the response of this patient vs the others? Include in discussion.

Reviewer #3 (Remarks to the Author): with expertise in clinical trial study design, biostatistics

General comments:

1. Additional details on the safety assessment should be included in the manuscript.
 - a. What is the starting dose? I assume it was 70 micro-grams but it was not explicitly stated.
 - b. What toxicities (and corresponding grades) are considered as dose-limiting-toxicities?
 - c. What are the criteria for a patient to be included in the "AE evaluable" population? What is the minimum dose that a patients need to receive to be included in the AE evaluable population?
 - d. What is the dose escalation scheme? It does not appear to be a 3+3. Is there a specific method that was used to determine whether to escalate the dose?
2. Can author state a recommended phase 2 dose given the totality of the data?

Specific comments:

1. The trial's eligibility criteria do not have restrictions on tumor type; however, all 26 patients enrolled had ovarian cancer. Is that a coincidence? Or it is because ovarian cancer patients tend to have higher expression of FR_alpha expression?
2. Page 6, line 133. Please include the allowable number of doses for maintenance period.
3. Table 2, the dose level table should be table 2 "e" rather than "d" as currently indicated.
4. Which CTCAE version was used for AE assessment? Page 8, line 164 mentioned CTCAE 5.0 but page 19, line 452 mentioned CTCAE 4.0.
5. CA125 was not mentioned in the methods section as a tool for evaluating anti-tumor activities. Some text should be added to explain the rationale of using CA125.
6. Can author elaborate why there are 4 patients who are not evaluable for efficacy (i.e. only 20 out of 24 were evaluable for efficacy)?

RESPONSES TO REVIEWER COMMENTS

'Safety and anti-tumour activity of MOv18 IgE, the first antibody therapy in a novel isotype class'

Spicer, J et al.

Remarks from the 3 reviewers are reproduced in full below, with responses from the authors after each point, highlighted in italics. Line numbers refer to the tracked change version of the manuscript:

Reviewer #1 (Remarks to the Author): with expertise in cancer clinical trials, folate receptor targeting

The manuscript presented the first clinical data that demonstrated the safety and effectiveness of an ADC using an IgE antibody for the treatment of ovarian cancer. The authors targeted the folate receptor alpha (FR α) for the clinical study and tested the IgE antibody candidate MOv18. The safety profile was quite acceptable. Basophil activation and signs of systemic toxicity were observed in only one out of the 26 patient participants and that patient showed baseline basophil activation prior to the treatment. In general, the ADC induced only low-grade, manageable adverse effects among the patients, and produced an elevation of serum IL-6, as well as MCP-1 and IFN γ levels post treatment.

On the aspect of anti-tumor activity, MOv18 treatment led to a marked reduction of serum CA125 in cycle 1 and 2, as well as a reduction of tumor mass in one of the 24 patients, though the marker reduction was not maintained beyond 28 days. The study offered insights into the potential of IgE class antibodies as an alternative to the existing IgG class antibodies in the treatment of solid tumors.

However, there are several points that may need to be clarified (listed below).

1. As indicated by the experimental design in Figure 2c, each patient received multiple infusions in the span of 10 weeks, including 6 weekly infusions and 2 biweekly infusions. However, the data that demonstrate the effect of Mov18 treatment in serum cytokine and tryptase levels are mostly showing only one cycle. Did the consecutive infusion cycles have any cumulative impact on the serum cytokines? Did the weekly and biweekly infusion cycles influence the serum cytokine levels differently?

Cytokine assays were performed at the timepoints indicated in Figure 3c, before and after the first dose. This time course spanned the first week of exposure to Mov18 IgE, but we did not collect further samples beyond these timepoints except in the case of the single case of intra-patient dose escalation.

Serum tryptase was measured in every patient at baseline, at the end of each infusion, and following any infusion-related event [line 566]. Tryptase was elevated only in one patient, at the time of their anaphylactic toxicity [line 187], and not in any other patient including those experiencing urticaria [line 188].

Did the adverse effects show any change in the subsequent infusion cycles? The authors may need to include a more thorough data/discussion that better explain the physiological impact of the treatment over the multiple infusion cycles.

The only common treatment-emergent adverse events were cutaneous manifestations of MOv18 IgE toxicity. In general, these were more prominent with first dose administration and became less obvious with subsequent doses, although this could have been a result of the use of secondary prophylaxis with steroid and antihistamine. This tachyphylaxis is mentioned in the 'Results, Safety' section [lines 175-176], and further exposition on this point has now been added to the Discussion [lines 389-392].

2. One out of all the participating patients showed marked reduction of the serum level of CA125 marker and reduction of the tumor mass. Are there any tumor or patient correlatives which explain the sensitivity to MOV18 therapy? Do we know the FR α expression? The authors should discuss the case in more details.

We agree that further details for this patient showing evidence of clinical benefit would be of interest. This patient was aged 62 at enrolment, and had a baseline performance status of 0. She was at least as heavily pre-treated as others recruited to this trial. The FR α expression in the tumour was 10%, amongst the lowest of patients recruited (range 10-95%). We have added some further comments to the 'Results, Anti-tumour activity section' [lines 312-315].

3. It might be informative to discuss more about how the performance of MOV18 compare to that of IgG class FR α antibodies in anti-tumor activities, and how the antibody may benefit the patients more over existing IgG class antibodies.

We have tested anti-tumour activity of IgE in comparison to IgG antibodies in pre-clinical models, as preparation for this trial (for example, Josephs, D et al. Cancer Res 2017, reference 14), demonstrating superior efficacy of IgE. This work is mentioned and cited in the Introduction [lines 81-84]. The implications of our trial result for creating a novel class of monoclonal antibody therapies, directed against tumour antigens already targeted by IgG drugs, are reviewed in the Discussion [lines 320-324, and 423-425].

There is not yet sufficient clinical data available to allow comparison of the activity of FR α -targeting IgE and IgG antibodies in humans.

Reviewer #2 (Remarks to the Author): with expertise in IgE biology, clinical

This is an important and interesting contribution. The phase I trial of an IgE to folate receptor-alpha showed anti-tumour activity of this antibody in a patient with ovarian cancer. The results are novel, of clinical significance, scientifically important, and well presented.

Major issues

None

Minor issues

The authors describe potential advantages of therapeutic IgEs of IgGs, i.e. 1) very high affinity

to FcεRI receptors, 2) expressed on monocytes/macrophages, mast cells, basophils and dendritic cells, 3) long retention of antibody by these effector cells, 4) absence of immune complex formation, 5) more potent immune response to cancer cells. They refer to Fig 1, which is nice but doesn't really show the how therapeutic IgE differ from IgGs. Consider adding a table that describes these differences (and similarities) and includes refs in their support.

We thank this reviewer for their suggestion and have added a new table as a panel in Figure 1, illustrating the key differences between IgG and IgE biology which are relevant in the therapeutic context. Supporting citations from the main text are also mentioned in the revised legend for this figure:

	IgG	IgE
Fc receptor affinity	+	+++
Tissue-resident effector cells	-	+
Duration of binding to effector cells	+	++
Immune complex formation	+	-

The authors performed skin prick testing and basophil testing with MOv18 IgE antibody to check for interaction with soluble factors, possibly originating from the target tumour. Why did the authors not check for the presence of targets of MOv18 IgE antibody in the blood of patients, e.g. by ELISA or WB?

In fact we did use ELISA to look for circulating soluble FRα and also for anti-FRα antibodies at a number of timepoints for each patient [‘Results’ lines 244-250; ‘Methods’ lines 587-593]. We emphasise in the ‘Discussion’ that serum concentrations of FRα and anti-FRα auto-antibodies were undetectable at all timepoints in the patient experiencing anaphylaxis [lines 372-373].

The authors say that patients potentially benefitting could continue with subsequent two-weekly maintenance dosing. How was this assessed? What were the criteria for "potential benefit"?

The protocol considered patients with stable disease or response by RECIST criteria at the first tumour evaluation (6 weeks) to be potentially benefitting from treatment. These patients were offered continued treatment. This clarification has been added to the ‘Results, Dose Escalation’ section [line 140].

The authors state that most common events were localised cutaneous toxicities including urticaria, pruritis and rash. All three terms are problematic. Urticaria is a disease (defined by the occurrence of itchy wheals - also called hives, angioedema, or both. Use of the word "wheals" or "hives" instead of "urticaria" is advised. Substitute wheal for urticaria throughout. Pruritis = pruritus. Rash is too vague, describe the specific skin reaction(s) that occurred.

'Urticaria', 'pruritus' and 'rash' are each classified in the Common Terminology Criteria for Adverse Events (CTCAE), Version 4 of which was selected in the trial protocol [lines 173, 478] for the description of adverse events in this trial:

https://evs.nci.nih.gov/ftp1/CTCAE/CTCAE_4.03/CTCAE_4.03_2010-06-14_QuickReference_8.5x11.pdf

The CTCAE, published and regularly updated by the US National Cancer Institute, are very widely used to record adverse events in oncology trials.

Each of 'urticaria', 'pruritus' and 'rash' is also a MedDRA preferred term (www.meddra.org). 'Rash' is not a MedDRA Lowest Level term, and the authors agree with this referee that this toxicity would ideally have been more specifically categorised in some toxicity reporting from this trial. However, these are the terms that were assigned by the treating investigator and recorded in the trial database for each event.

Urticaria is defined in the CTCAE as, 'A disorder characterized by an itchy skin eruption characterized by wheals with pale interiors and well-defined red margins'. This description accurately captures the appearance of the skin eruption seen in many patients receiving MOv18 IgE. Furthermore, urticaria is used as part of the definition and grading of anaphylaxis in the CTCAE, and in the 2020 World Allergy Organisation anaphylaxis guideline. The authors (including allergists CC and ST) feel this terminology will most unambiguously describe the clinical picture associated with MOv18 IgE.

We have corrected the spelling of pruritus in the text [p.8].

The antibody was administered as an intravenous infusion. Please add volume and infusion duration.

The first two doses of MOv18 IgE were administered as an intravenous infusion in 250ml 0.9% saline over 90 minutes. A shorter infusion time for subsequent doses of 60 minutes was allowed if no adverse reactions are encountered. This information has been added to the Methods [lines 505-507].

Control IgE antibodies were used for basophil testing. Which ones?

The control IgEs were non-FR α reactive IgE antibodies, now stated in the relevant Methods section [lines 553-554].

What were the patients' total IgE levels? Were these associated with efficacy and safety?

Patients' endogenous IgE levels varied widely, as is commonly seen in human populations. There was no correlation with observed toxicity or anti-tumour activity, as shown in the following Figure (which we now include as Supplementary Figure 2):

Legend: Light blue fill = patients experiencing urticaria; dark blue outline = patient experiencing clinical benefit

Did any of the patients have anti-IgE Abs?

We looked for the emergence of anti-drug antibodies (ADA, specific to MOv18 IgE) only, mentioned in 'Results, Anti-drug antibody responses' (lines 268-271). The figure below shows that these were detected at any time point in only 3 patients. ADA were detected in 2 of 8 patients in Cohort 2 (one patient at 28-day follow up, and another at infusion 6 and 28-day follow up [*]), and in 1 of 6 patients in Cohort 3 (at 28-day follow up [†]). There was no correlation between ADA titre and toxicity or clinical benefit. A summary of ADA detection amongst patients in this trial is now included as a new Supplementary Figure 1:

In line 370, the authors refer to urticaria: The most common toxicity of MOv18 IgE was urticaria. Although the extent of this rash varied greatly between patients, it was tolerable and transient.

Please see my comment above. Urticaria is vague, and rash is even more vague. What signs and symptoms did patients develop? Itchy wheals? If yes, when, how many, where, duration? Angioedema? If yes, when, where, duration? Rash is too unspecific. Avoid the use of this term. *'Urticaria' in our opinion describes a quite specific clinical sign, and our co-authors with long experience in the clinical practice of allergy support this statement, as discussed above. Once again, we agree that 'rash' could have been more specifically categorised in some cases. Informal discussion with investigators across the participating clinical sites suggests that 'rash' was used as a descriptor in the context of a mucular rash, which would therefore be distinct from urticaria (characterised by wheals, by definition a papular eruption). To reflect these qualitative comments and to address this reviewer's appropriate request for more information about these cutaneous toxicities, we have added text to the Results section [line 385-392].*

Only one patient had clinical benefit. How was this patient different than the others? What may be the explanation for the response of this patient vs the others? Include in discussion. *This patient did not have any obvious characteristics that might account for clinical benefit, where this was not seen in others. For example, they had not received fewer prior therapies [lines 310-312]. Furthermore, membrane FR α expression in their tumour was 10%, where the range in the treated population was 5-95% [Figure 2b]. We have now included discussion of this [lines 312-315].*

Reviewer #3 (Remarks to the Author): with expertise in clinical trial study design, biostatistics

General comments:

1. Additional details on the safety assessment should be included in the manuscript:
 - a. What is the starting dose? I assume it was 70 micro-grams but it was not explicitly stated. *It is correct to assume the starting dose was 70ug, and this has now been explicitly stated [lines 139, 523].*
 - b. What toxicities (and corresponding grades) are considered as dose-limiting-toxicities? *Dose limiting toxicities are defined on p.28 of the study protocol. DLT was defined as any grade 3/4 non-haematological adverse event (with exceptions in the case of incomplete supportive medication for diarrhoea or vomiting, grade 3 fatigue, or biochemical abnormalities deemed clinically non-significant). Haematological DLTs were defined as grade 4 neutropenia or thrombocytopenia that was prolonged, or grade 3/4 neutropenia associated with fever or infection. This information has been added to the Methods section [lines 529-534].*
 - c. What are the criteria for a patient to be included in the "AE evaluable" population? What is

the minimum dose that a patients need to receive to be included in the AE evaluable population?

All patients receiving at least 1 intravenous dose of MOv18 IgE were evaluable for safety [see Figure 2 legend].

d. What is the dose escalation scheme? It does not appear to be a 3+3. Is there a specific method that was used to determine whether to escalate the dose?

The dose escalation scheme is presented on page 37 of the trial protocol, and the following summary has been added to the manuscript Methods section [lines 518-522]. An accelerated dose escalation scheme, starting with single-patient cohorts, was planned up to cohort 5 (1.5mg). An additional 2-5 patients could be added to each cohort in the event of emergent toxicity. Thereafter, three patients were to be enrolled per cohort, with an additional three patients added to a cohort if needed for toxicity.

2. Can author state a recommended phase 2 dose given the totality of the data?

We state in the abstract [line 60] that maximum tolerated dose was not reached. But we have safely delivered this drug at doses much higher than associated with signs of anti-tumour activity. A company (Epsilogen Ltd) has been founded and has successfully raised new funding to deliver a follow-on Phase 1b/2 trial, to further explore dose and efficacy of MOv18 IgE.

Specific comments:

1. The trial's eligibility criteria do not have restrictions on tumor type; however, all 26 patients enrolled had ovarian cancer. Is that a coincidence? Or it is because ovarian cancer patients tend to have higher expression of FR_alpha expression?

We did screen a range of solid tumours for FR α expression. Based on expectations from the literature these included mesothelioma and endometrial cancers. However, ovarian and to a lesser extent endometrial cancers were the only tumour types in which we found a significant prevalence of antigen expression. This may have been because of the careful validation of the IHC assay performance that was required before its use in this trial, which could account for higher specificity than alternative assays previously reported. A sentence explaining this has been added to 'Results' [lines 135-136].

2. Page 6, line 133. Please include the allowable number of doses for maintenance period. *Maintenance allowed a further three two-weekly doses. This information has been added [line 141 in the revised manuscript].*

3. Table 2, the dose level table should be table 2 "e" rather than "d" as currently indicated. *Thank you, this has now been corrected.*

4. Which CTCAE version was used for AE assessment? Page 8, line 164 mentioned CTCAE 5.0 but page 19, line 452 mentioned CTCAE 4.0.

This trial used CTCAE Version 4.0, now corrected [line 173 in the revised manuscript].

5. CA125 was not mentioned in the methods section as a tool for evaluating anti-tumor activities. Some text should be added to explain the rationale of using CA125.
This has now been added in 'Methods' [lines 460-461].

6. Can author elaborate why there are 4 patients who are not evaluable for efficacy (i.e. only 20 out of 24 were evaluable for efficacy)?
Cohort 2 included 4 patients not evaluable for safety who did not receive any intravenous dosing because of positive intradermal tests [Figure 2 legend].

Response to journal flow cytometry reporting requirements:

Flow gating strategy is summarised in a new Supplementary Figure 3. Gating strategy aimed to identify $CCR3^{high}SSC^{low}$ basophils in unfractionated whole blood samples. Representative plots are shown of patient blood incubated with positive control stimuli (PC1 = anti-FcεRI, PC2 = fMLP, PC3 = anti-IgE), MOv18 IgE, or control IgE [detailed at lines 552-554]:

REVIEWER COMMENTS

Reviewer #1 (Remarks to the Author):

The authors have adequately responded to the criticisms.

Reviewer #2 (Remarks to the Author):

The authors comprehensively addressed my concerns

Reviewer #3 (Remarks to the Author):

Major comments:

1. The number of patients does not seem to match up. The manuscript states that 27 patients were enrolled, 26 were eligible, and 24 were treated (i.e. 2 eligible patients were not treated).
 - a. Figure 2 legend states that "cohort 2 included 4 patients not evaluable for safety who did not receive intravenous dosing because of positive intradermal tests". Figure 2e has cohort 2 as "6 (No. patients treated) / 8 (No. enrolled)" which tells me there are only 2 patients in cohort 2 who were not treated. So, how many people exactly not treated in cohort 2?
 - b. There are 20 patients who are evaluable for efficacy and the previous comment asked for the reason of attribution (from 24 treated to 20 evaluable for efficacy). Authors responded to the previous comment that "Cohort 2 included 4 patients not evaluable for safety who did not receive any intravenous dosing because of positive intradermal tests" which implies that the safety population and efficacy population are the same. If that is the case, only 20 patients in total were treated but the manuscript states 24.
 - c. It is fine for a manuscript to have different analysis populations (i.e. safety evaluable population and efficacy population in this manuscript) but it is important to provide the definition of the different populations. It is also important to provide enough information so the reviewers can understand how you reach the number of patients in each population.

Minor comments:

1. The dose escalation scheme is atypical. Do authors have a reference paper for this dose escalation scheme that can be included in the manuscript? If no reference can be provided, please justify the dose escalation scheme.
2. Thank you for pointing out that the AE evaluable definition is in the Figure 2 legend. AE evaluable definition is important for trial design; therefore, please also state your AE evaluable definition in the method section.
3. Please state the AE evaluable period in the method section. For example, does DLT only take into account the AEs occurred after the first dose?

Reviewer #3 (Remarks to the Author):

We thank this reviewer for their careful scrutiny of the manuscript, and provide the following clarifications and corrections. Authors' responses are in red:

Major comments:

1. The number of patients does not seem to match up. The manuscript states that 27 patients were enrolled, 26 were eligible, and 24 were treated (i.e 2 eligible patients were not treated).

Throughout the manuscript we consistently state that the total number of patients recruited was 26. The number 27 may arise from misinterpretation of the patients-per-cohort shown in Figure 2(e), where the same individual appears both in cohort 7(i) and in cohort 7(ii). This patient was the subject of intra-patient dose escalation, as indicated by [†] in the legend to that Figure. It is correct that 2 of these 26 patients recruited did not go on to be treated with IV drug, as emphasised below.

a. Figure 2 legend stat that "cohort 2 included 4 patients not evaluable for safety who did not receive intravenous dosing because of positive intradermal tests". Figure 2e has cohort 2 as "6 (No. patients treated) /8 (No. enrolled)" which tells me there are only 2 patients in cohort 2 who were not treated. So, how many people exactly not treated in cohort 2?

The reviewer is correct. The numbers in the Figure 2(e) panel are accurate, with 6 patients treated out of 8 enrolled in cohort 2. However, the legend erroneously stated that 4, rather than 2, patients received no intravenous dose of MOv18 IgE because of a positive intradermal test at baseline. The number evaluable for safety is therefore $26 - 2 = 24$. We thank the reviewer for pointing out this inconsistency, and have corrected the Figure 2 legend accordingly (page 7).

b. There are 20 patients who are evaluable for efficacy and the previous comment asked for the reason of attribution (from 24 treated to 20 evaluable for efficacy). Authors responded to the previous comment that "Cohort 2 included 4 patients not evaluable for safety who did not receive any intravenous dosing because of positive intradermal tests" which implies that the safety population and efficacy population are the same. If that is the case, only 20 patients in total were treated but the manuscript states 24.

Patients were only evaluable for efficacy if they had at least one on-treatment CT scan.

We apologise again for the error discussed in response to point (a) above, which also addresses the question of the number of patients evaluable for safety. The Figure 2(e) legend has been corrected, as already stated, and now reads, "Cohort 2 included 2 patients not evaluable for safety who did not receive any intravenous dosing because of positive intradermal tests". This error arose because a total of 4 patients withdrew because of positive intradermal tests, but 2 of these had by then received an IV dose of MOv18 IgE, and were therefore evaluable for safety. The number of patients treated with IV drug, and therefore evaluable for safety, is 26 (enrolled) less 2 (positive intradermal test at baseline), for a total of 24.

c. It is fine for a manuscript to have different analysis populations (i.e. safety evaluable population and efficacy population in this manuscript) but it is important to provide the definition of the different populations. It is also important to provide enough information so the reviewers can understand how you reach the number of patients in each populations.

We hope the above explanations adequately account for the stated numbers of patients evaluable for safety and efficacy.

Minor comments:

1. The dose escalation scheme is atypical. Do authors have a reference paper for this dose escalation scheme that can be included in the manuscript? If no reference can be provided, please justify the dose escalation scheme.

After a conservative starting dose of 70 μ g, an approximate and conventional doubling of doses in subsequent cohorts was planned (250 μ g, 500 μ g, 700 μ g, 1.5mg, 3mg, 6mg, 12mg). Escalation using single patient cohorts, until significant toxicity was observed or a dose of 1.5mg was reached, is described in Methods (page 21). This scheme is broadly accepted and employed in cancer trials when a there is suspicion that a starting dose may be well below the expected range of biological activity (see for example Simon et al. (1997) Accelerated titration designs for Phase I clinical trials in oncology. J Natl Cancer Inst 89:1138). This citation can be included if required.

2. Thank you for pointing out that the AE evaluable definition is in the Figure 2 legend. AE evaluable definition is important for trial design; therefore, please also state your AE evaluable definition in the method section.

We have added the following statement to the Methods (page 19), "All patients receiving at least 1 intravenous dose of MOv18 IgE were evaluable for safety."

3. Please state the AE evaluable period in the method section. For example, does DLT only take into account the AEs occurred after the first dose?

The DLT period was three weeks, and we have added this important information to the Methods (page 21), "DLT was defined as ... occurring in the first 3 weeks of MOv18 IgE administration."

REVIEWERS' COMMENTS

Reviewer #3 (Remarks to the Author):

Thank you for addressing my comments.

Please kindly do the following editorial changes:

1. Line 289, Please change the sentence to "Twenty of 24 patients, who had at least one on-treatment CT scan, were evaluable for efficacy (Figure 5a)."
2. Around Line 510. Please include the reference of Simons (1997) for the dose escalation scheme.

Reviewer #3

Thank you for addressing my comments.

Please kindly do the following editorial changes:

1. Line 289, Please change the sentence to "Twenty of 24 patients, who had at least one on-treatment CT scan, were evaluable for efficacy (Figure 5a)."
2. Around Line 510. Please include the reference of Simons (1997) for the dose escalation scheme.

These changes are made.